# Spontaneous verbal descriptions of vegans, non-vegan vegetarians, and omnivores and relationships between these descriptions and perceivers' diets

John B. Nezlek[1,2☯]*, Catherine A. Forestell[2☯], Harini Krishnamurti[3]

1 Center for Climate Action and Social Transformations (4CAST) Institute of Psychology, SWPS University, Warsaw, Poland, 2 Department of Psychological Sciences, College of William & Mary, Williamsburg, Virginia, United States of America, 3 Department of Psychological and Brain Sciences, George Washington University, Washington, DC, United States of America

☯ These authors contributed equally to this work.
* jbnezl@wm.edu

**Data Availability Statement:** All raw data, including a codebook, are available via the OSF data repository: https://doi.org/10.17605/OSF.IO/2XD8M.

## Abstract

Participants, 672 US collegians, listed four words/terms that first came to mind when thinking of vegans, non-vegan vegetarians, and omnivores. Participants generated 1264 unique descriptors, which two sets of raters, who were blind to the source of the descriptors, rated on 10 dimensions that included the valence of the descriptors (i.e., positive, negative). A series of multilevel models in which descriptors were nested within persons, found that descriptors referring to environmental issues and health were used more frequently when describing both vegans and vegetarians than when describing omnivores. Descriptors referring to deviance, lifestyle, and politics were used more frequently when describing both vegans and vegetarians than when describing omnivores. Overall, vegans were viewed more negatively than vegetarians who were viewed more negatively than omnivores. These differences were moderated by the extent to which participants restricted meat from their diet. Those who restricted meat from their diets to a greater extent had more negative perceptions and fewer positive perceptions of omnivores, whereas they had more positive perceptions of vegans and vegetarians, and fewer positive perceptions of omnivores. The present study is the first to use spontaneous verbal reports to examine attitudes and perceptions of people based on their eating habits. The results suggest that dietary habits can serve as a basis for social identity, which in turn affects perceptions of others.

## Introduction

Strictly speaking, vegetarianism is a dietary habit in which the consumption of animal-based foods, such as meat and fish, is avoided. Although defined in terms of dietary habit, vegetarianism is considered to be more than a set of dietary preferences—rather it reflects one's social identity. According to Social Identity Theory, when people categorize themselves and others

**Funding:** This research was supported by a grant to John B. Nezlek, 2018/31/B/HS6/02822 from the Narodowe Centrum Nauki (Polish National Science Centre), https://www.ncn.gov.pl/. The funders had no role in study design, data collection and analysis, decision to publish, or preparation of the manuscript.

**Competing interests:** The authors have declared that no competing interests exist.

into groups, they tend to have ideas, opinions, knowledge, and beliefs about specific social objects that are similar to other members of their in-groups [1]. Because food choice is a domain within which people can readily express their ideals and identities, it serves to connect communities of individuals who share similar food-choice patterns and by extension, similar values and beliefs [2].

Consistent with this, food choices may come to represent an individual's broader life philosophy and can become entwined with other aspects of self-concept. For example, the available data suggest that vegetarians are more prosocial people than omnivores. This has been demonstrated by research showing that vegetarians are more empathetic toward humans and animals [3,4] and more altruistic, advocating for values such as protecting the environment, animal welfare, equality, and social justice more strongly than omnivores [5]. In contrast, omnivores tend to be more authoritarian, higher in social dominance orientation [6–8] and more politically conservative than vegetarians [9]. See Nezlek and Forestell [10] for a further discussion and Rosenfield and Burrow [11] for an example of a multicomponent model of vegetarian identity.

Social identity theory also posits that when people categorize themselves into groups, they tend to think of their in-group as better than and as more heterogeneous than out-groups [1]. Because the overwhelming majority of people are omnivores, at least in Western industrialized nations such as the US [e.g., 12], vegans and vegetarians (hereafter collectively referred to as veg*ns) are a social minority. As such, they may be subject to some of the same types of treatment that other social minorities experience, including derogation and disrespect.

## Previous research on perceptions of people as a function of their diets: Substantive conclusions

Consistent with conceptualizing veg*ns as a social minority, previous research has focused primarily on how omnivores view veg*ns. Although the results are somewhat mixed, on balance, veg*ns tend to be viewed less positively or more negatively than omnivores and vegans tend to be viewed more negatively than non-vegan vegetarians [e..g., 13]. For example, in a frequently cited study, based on feeling thermometers, MacInnis and Hodson [14] found that omnivores evaluated veg*ns as a whole less positively than they evaluated other omnivores, and they evaluated vegans less positively than they evaluated non-vegan vegetarians (hereafter referred to as vegetarians).

In contrast, in a sample of mostly female, liberal, American university students, Chin, Fisak, and Sims [15] found that attitudes toward vegetarians were generally positive. This finding was replicated more recently in a sample of adults from New Zealand. In this study, Judge and Wilson [16] found that omnivores generally evaluated veg*ns positively, although vegans were evaluated less positively than vegetarians. According to Ruby et al., [17] attitudes toward vegetarians may differ cross culturally. In this study, although Argentinian, Brazilian, French, and American students' attitudes toward vegetarians were relatively neutral, admiration of vegetarians was highest in Brazil and the USA and French participants were "bothered" most by vegetarians. It is worth noting that in all of these studies, omnivorous participants did not evaluate other omnivores, making it impossible to determine how veg*ns were perceived relative to omnivores. Studies that have compared people's perceptions of vegetarians and omnivores have reported that while people perceive vegetarians to be more virtuous and moral than omnivores, they also perceive them to be more feminine, weak, and less masculine [18,19]. Regardless of omnivores' reported attitudes and perceptions, veg*ns believe that others treat them more negatively as a result of their diet. For example, Nezlek et al. [20] found that compared to omnivores, veg*ns reported that others treated them more negatively because of their

diets. Interestingly, Nezlek et al. also found that compared to omnivores, veg*ns reported that others treated them more positively in some ways because of their diets (e.g., veg*ns reported that they were admired more, and others treated them in a friendly way relative to omnivores). Nevertheless, on balance, perceptions of negative treatment were stronger and occurred more regularly than perceptions of positive treatment across the three studies Nezlek at al. conducted.

## Previous research on perceptions of people as a function of their diets: Types of measures

The majority of research on how people perceive veg*ns has relied upon what are referred to as "closed-ended" questions. Participants respond to statements using scales provided by a researcher [e.g., 15]. Closed-ended questions require respondents to provide evaluations using the frameworks provided by researchers, which may or may not correspond to or represent how the respondents themselves construct their evaluations. Although the existing research on how people evaluate others as a function of a target's diet has provided some important insights, we think it is valuable to complement the existing research on perceptions of veg*ns by studying spontaneous verbal descriptions of others.

The use of verbal descriptions as a way to understand what people are thinking, including how they evaluate others, has a long history in psychology. Spontaneous verbal descriptions may represent people's thoughts more clearly than responses to closed-ended questions that limit respondents to provide evaluations within contexts prescribed by researchers. We are aware of only one study, Study 1 in Minson and Monin [18], that used open-ended response formats to study perceptions of vegetarians. In this study, close to half of omnivores generated at least one negative word to describe vegetarians. Unfortunately, Minson and Monin did not collect open-ended responses describing omnivores, so it is not clear whether the 47 undergraduates (all omnivores) in their study viewed vegetarians more or less positively or negatively than they viewed omnivores.

## The present study

To complement and extend the existing research on how diet is related to person perception, in the present study participants described their dietary habits, and provided spontaneous verbal descriptions of vegans, vegetarians, and omnivores. We rated these descriptions on 10 dimensions suggested by previous research on person perception and on perceptions of vegetarians.

## Dimensions used to code descriptors

Based on the classic work by Osgood et al. [21] on the semantic differential, we coded for the valence of the descriptors; i.e., did descriptors refer to something positive and did they refer to something negative. We chose positive and negative because Osgood et al. found that valence was the dominant organizing theme for a variety of judgments and evaluations. We did not define positive and negative as polar opposites of a single dimension; rather, we defined them as two separate dimensions.

We also coded for the three motives that people have described as the reasons for reducing their consumption of meat: concerns about the ethics of meat production (e.g., animal welfare), concerns about the impact of meat production on the environment, and beliefs that eating meat is not healthy [22,23].

Following suggestions that vegetarianism is a social identity [10], we measured this construct broadly, as it might be expressed by lay people, and we coded descriptors for whether

they referred to lifestyle. In addition, based on previous research demonstrating that vegetarians hold more liberal values and voting habits than omnivores [24], we coded descriptors in terms of whether they referred to politics. Previous research has found that eating meat tends to be associated with masculinity [19,25], and given that we were primarily interested in perceptions of veg*ns, we coded descriptors for femininity or a lack of masculinity.

Although point estimates vary, vegetarians are clearly a social minority in Western, industrialized countries. The study was conducted in the US, where the percent of vegetarians has been about 5% for the past 20+ years [12]. Given the minority status of vegetarians, we coded descriptors for whether they referred to deviance or minority status. Finally, given that vegetarianism is defined in terms of the types of food people consume (or avoid consuming), we coded descriptors based in terms of whether they referred to food. Descriptions of how these dimensions were defined are provided in the methods section.

## Present study: Hypotheses and expectations

Although research on perceptions of veg*ns does not uniformly find that veg*ns are perceived negatively, given that veg*ns are a social minority, we expected that negative descriptors would be used more frequently when describing veg*ns (vegans and vegetarians) than when describing omnivores. Furthermore, we expected that negative descriptors would be used more frequently when describing vegans than when describing vegetarians. Given the mixed findings regarding positive evaluations of veg*ns, we had no clear expectations regarding differences in how often positive descriptors would be used.

Given that animal rights, concerns about the environment, and health concerns are the three primary reasons why people adopt veg*n diets [26], we expected that such descriptors would be used more frequently when describing veg*ns than when describing omnivores. In terms of differences in the descriptions of vegans and vegetarians, although we had no clear expectations regarding such differences in the use of these three classes of descriptors, we examined them.

Given the likely existence of vegetarianism as a social identity [10], including politics and lifestyle [24], we expected that descriptors reflecting lifestyle and politics would be used more frequently when describing veg*ns than when describing omnivores, and that descriptors related to politics would be used more often when describing vegans than when describing vegetarians. Moreover, given the fact that consuming meat is associated with masculinity and veg*n diets are associated with femininity [25], we expected that descriptors associated with stereotypical feminine traits would be used more frequently to describe veg*ns than to describe omnivores. We had no clear expectations about whether how frequently feminine descriptors would be used when describing vegans and vegetarians; nevertheless, we examined such differences.

Given the minority status of veg*ns, we expected descriptors reflecting deviance would be used more frequently when describing veg*ns than when describing omnivores, and that deviance descriptors would be used more often when describing vegans than when describing vegetarians. This second expectation was based on the fact that vegans are often perceived as more extreme (deviant) than vegetarians [e.g., 13].

We also examined relationships between participants' diets and their use of descriptors for the three targets. Consistent with the well-established principle that similar others are viewed more positively [e.g., 27], we expected that for veg*n targets, the use of positive descriptors would be positively related to how much participants restricted meat consumption, whereas for omnivore targets the use of positive descriptors would be negatively related to how much participants restricted meat consumption. We expected the opposite relationships for how

often negative descriptors were used. We expected that for veg*n targets, the use of negative descriptors would be negatively related to how much participants restricted meat consumption, whereas for omnivore targets the use of negative descriptors would be positively related to how much participants restricted meat consumption.

There was no basis to form clear expectations regarding relationships between participants' meat restriction and the use of the other descriptors because they did not have an explicit positive or negative valence. Nevertheless, we examined such relationships on an exploratory basis.

## Method

### Participants

Participants were undergraduate students at a US university who participated in partial fulfillment of a course requirement ($N = 672$; $M_{age}$ = 18.9 years, $SD$ = 1.24; 418 women, 239 men, 15 other). The study was conducted between September and December, 2020.

### Ethical statement

The study was conducted in accordance with the Declaration of Helsinki regarding the rights of research participants. Participants provided explicit consent electronically. Participants were told the purpose of the study and were told that they could decline to answer any question without penalty. Responses were de-identified prior to analysis. The study was approved on 9/10/2020 by the Protection of Human Subjects Committee, College of William & Mary, protocol: PHSC-2020-08-12-14441-tmthra.

### Measures

Participants described their dietary habits using a measure developed by Forestell, et al. [28]. Participants indicated if they were vegan, someone who eats fruits, vegetables, and grains but no animal or seafood products ($n = 10$); lacto-vegetarian, someone who eats fruits, vegetables, grains, and dairy products, but no other animal or seafood products ($n = 4$); lacto-ovo-vegetarian, someone who eats fruits, vegetables, grains, dairy products, and eggs, but no other animal or seafood products ($n = 26$); pescetarian, someone who eats fruits, vegetables, grains, dairy products, eggs and seafood, but no other animal or seafood products ($n = 20$); semi-vegetarian, someone who eats fruits, vegetables, grains, dairy products, eggs, seafood, and poultry but no red meat ($n = 43$); occasional omnivore, someone who occasionally eats red meat, poultry, seafood, eggs, dairy products, fruits, vegetables, and grains ($n = 86$); or omnivore, someone who regularly eats most meats, seafood, eggs, dairy products, fruits, vegetables, and grains ($n = 480$). Three participants choose not to answer this question.

Participants were then asked to describe what they thought of vegans, vegetarians, and omnivores. They responded to three separate prompts: "What four words come to mind when you think of vegans (vegetarians/omnivores)?" Participants entered their descriptions as text. There were no other suggestions or instructions.

### Coding of descriptors

First, a list of unique descriptors was created. After eliminating duplicates due to pluralization (e.g., females and female), misspellings, and alternate spellings (e.g., life style and life-style) there were 1264 descriptors. Synonyms were not considered as duplicates nor were descriptors that had common roots (e.g., environmentalism and environmentalist). An overview of the dimensions used for coding and examples for each are provided in Table 1, and a list of the descriptors that were rated is presented in the supplemental materials. The mean number of

**Table 1. Descriptions of dimensions with examples.**

| Category | Dimension | Definition | Examples |
|---|---|---|---|
| Valence | 1. Positive | Referring to or reflecting something positive about the person. | admirable; aware; moralsmart; strong |
| | 2. Negative | Referring to or reflecting something negative about the person. | elitist; obnoxious; picky; unaware; wasteful; weak |
| Motives for restricting meat intake | 3. Health | Referring to symptoms of health or to health-related behaviors | healthy; vitamins; malnourished; lean; |
| | 4. Environment | Referring to behaviors or ideas associated with protecting (or destroying) the environment | eco-friendly; environment; sustainable |
| | 5. Animal Rights | Referring to behaviors or ideas associated with protecting or caring for animals | animal protection; inhumane; PETA[a]; save the animals; slaughter |
| Social Identity | 6. Lifestyle | Referring to the typical way of life of an individual, group, or culture, which can reflect a set of beliefs and habits. Explicitly distinguished from personality traits | American;hippie; lifestyle; rich; traditional |
| | 7. Politics | Referring to a political party, a politically motivated belief system, or a political ideology | liberal; activist; progressive; woke[b] |
| | 8. Feminine | Referring to characteristics or behaviors associated with traditional stereotypes about women/men | caring; flowery; girl; soft; women |
| | 9. Deviance | Referring to or reflecting behaviors that are maladaptive or harmful to oneself or to others or referring to being an outcast or not fitting in. | crazy;extreme; ignorant; weird; |
| Other | 10. Food | Referring to a food item or food in general | beans, carrot; cow;; dairy; eggs; fruit; meat;kale |

Notes

[a] refers to People for the Ethical Treatment of Animals; an American nonprofit animal rights organization

[b] defined as being aware of and actively attentive to important especially issues of racial and social justice.

descriptors participants generated was 9.95 ($SD$ = 2.96), and 75% of participants generated 8 or more descriptors.

The first set of ratings was done by five undergraduate research assistants. We used undergraduates because we thought they would be familiar with the participants' linguistic subculture. Each assistant rated each descriptor on 10 dimensions (yes/no ratings), which are listed and defined in Table 1. These independent ratings served as a basis for the final evaluation of two co-authors, both faculty members. A consensus coding procedure was used in which each co-author independently examined the ratings of the research assistants and made a final judgment. These two sets of ratings were then compared, and any discrepancies were resolved. To be scored 1, both final raters needed to agree that a descriptor referred to a certain dimension. Ratings of descriptors were completely blind. Neither the original raters nor the two final raters knew to which target a descriptor referred, nor did the raters know who generated the descriptors.

This procedure resulted in ten binary (0, 1) scores for each descriptor. The mean number of category scores the descriptors received was 1.13 ($SD$ = .71). Approximately one quarter of descriptors (24%) had no dimension scores, and 59% of descriptors referred to only one dimension. Our analyses included all the descriptors participants generated. All raw data, including a codebook, are available via the OSF data repository: https://doi.org/10.17605/OSF.IO/2XD8M.

## Results

### Overview of analyses

We conceptualized the data as a multilevel data set, with descriptors (words) nested within persons, and we analyzed the data using HLM [V. 8.23, 29]. Multilevel modeling (MLM) is needed to analyze what are sometimes referred to as nested or hierarchically nested data

structures. Nested data cannot be analyzed using single-level regression because nested data violate one of the foundational assumptions of regression–the independence of errors of observation. In the present study, individual participants generated multiple descriptors. This meant that descriptors could not be treated as independent observations.

MLM analyzes phenomena at multiple levels of analysis simultaneously. In the present study, this meant separate models at the word-level (descriptor) and at the person-level. For an introduction to MLM see Nezlek [30].

Whether a descriptor was associated with a characteristic was the outcome. This was defined in terms of binomial variable (yes/no), and so we used logistical multilevel regression. Logistical regression was needed because by definition, the mean and variance of the outcome were not independence, and such independence is a critical assumption of linear regression analyses.

The basic formula is below. In this model, there are $i$ words (descriptors) nested within $j$ participants/persons. A log-odds, $\eta_{ij}$, is estimated for each person. The random effect of the mean (the person-level variance, how much the log-odds for people, varied), is estimated by the variance of $\mu_{0j}$. Note that, because the outcome is a binomial, by definition, there is no level-1 (word-level) variance.

Word-level (level 1): $\text{Prob}(\text{Categ}_{ij} = 1 | \beta_j) = \varphi_{ij}$

$\log[\varphi_{ij} / (1 - \varphi_{ij})] = \eta_{ij}$

$\eta_{ij} = \beta_{0j}$

Person-level (level 2): $\beta_{0j} = \gamma_{00} + \mu_{0j}$

As explained below, the frequencies with which descriptors were associated with vegans, vegetarians, and omnivores were examined by adding predictors to the word-level (level-1) model shown above. Relationships between these frequencies and the extent to which participants restricted meat from their diets were examined by adding predictors to the person-level (level 2) model that was used to examine differences in evaluations of vegans, vegetarians, and omnivores.

## Descriptive statistics

The first analyses were "totally unconditional models," i.e., no predictors at either level of analysis, which is the basic model presented above. These analyses estimated the sample mean and the person-level variance, and these estimates are presented in Table 2. Although MLM analyses do not estimate level-1 (word-level) variances for binomial outcomes, they can estimate confidence intervals for the mean, and these are presented in Table 2. Note that the analyses

Table 2. Estimates of mean percent of words receiving different evaluations, 95% confidence intervals, and person-level variances.

| | | Confidence Intervals | | |
| --- | --- | --- | --- | --- |
| | Percent | Lower | Upper | Variance |
| Positive | 26.91 | 24.81 | 29.13 | 1.41 |
| Negative | 12.08 | 11.03 | 13.19 | 0.78 |
| Health | 11.77 | 10.79 | 12.82 | 0.66 |
| Animal rights | 0.57 | 0.40 | 0.79 | 2.39 |
| Environment | 7.54 | 6.80 | 8.34 | 0.54 |
| Lifestyle | 6.69 | 6.02 | 7.49 | 0.72 |
| Politics | 0.86 | 0.70 | 1.09 | 0.12 |
| Feminine | 1.41 | 1.19 | 1.77 | 0.83 |
| Deviance | 1.08 | 0.89 | 1.38 | 0.40 |
| Food | 26.20 | 24.01 | 28.52 | 1.64 |

estimate log-odds, which can then be converted into percentages. The variance estimates represent how much the estimated log-odds for people vary.

As can be seen from the mean estimates, some characteristics were assigned to less than 5% of descriptors. Such low baseline rates can make it difficult to analyze variance. Nevertheless, these are overall rates, which are collapsed across the descriptors generated for all three targets (vegans, vegetarians, and omnivores). Also, these estimates do not take into account participants' meat restriction. As described below, some of these relatively infrequently occurring characteristics varied as a function of both the diet of the person being described and the extent to which participants restricted meat from their diets.

## Differences among descriptors of vegans, vegetarians, and omnivores

Differences in the frequency with which the use of descriptors varied as a function of the diet of the target were examined with a series of analyses that included three level-1 predictors, one representing each diet. These predictors were dummy-coded (0, 1), and they were entered uncentered. The intercept was dropped, resulting in the model presented below. Such a "no-intercept" model estimates a mean log-odds of the percent of times a word was used when describing a person following each type of diet. These log-odds can then be used to estimate mean percentages. The random effects for each of these coefficients (the person-level variances of the log-odds for the three coefficients) are the variances of $\mu_{1j}$, $\mu_{2j}$, and $\mu_{3j..}$ This type of analysis is described in Nezlek [31].

Word level (level 1): Prob(Categ$_{ij}$ = 1|$\beta_j$) = $\varphi_{ij}$

log[$\varphi_{ij}$ /(1 - $\varphi_{ij}$)] = $\eta_{ij}$

$\eta_{ij}$ = $\beta_{1j}$ (vegan) + $\beta_{2j}$ (vegetarian) + $\beta_{3j}$ (omnivore)

Person-level (level 2): $\beta_{1j}$ = $\gamma_{10}$ + $\mu_{1j}$

$\beta_{2j}$ = $\gamma_{20}$ + $\mu_{2j}$

$\beta_{3j}$ = $\gamma_{30}$ + $\mu_{3j}$

The coefficients representing the mean log-odds for the diets of the targets were compared with a series of constraints on the model, e.g., constraining the coefficients for vegans ($\gamma_{10}$) and omnivores ($\gamma_{30}$) to be the same. Constraints were tested by comparing the goodness of fit for a model that included the constraint to the goodness of fit of a model that did not include the constraint. The difference between the two fit measures is distributed as a chi-square with 1 *df*. If the chi-squared was significantly different from 0, the constraint reduced the model fit, and the coefficients were not the same.

In most analyses, the random error terms for all three effects were not significant (tests of the variances of $\mu_{1j}$, $\mu_{2j}$, and $\mu_{3j}$). Nevertheless, fixing these effects, i.e., not estimating a random effect, did not meaningfully change the estimates coefficients nor the results of testing the constraints of the fixed effects, the $\gamma_{10}$, $\gamma_{20}$, and $\gamma_{30}$, coefficients, which are the focus of this paper.

The estimated percent of each characteristic for each type of target (transformations of the log-odds) and the results of the tests of the constraints of the equality of the coefficients used to estimate these means are presented in Table 3. A detailed summary of the results of the tests of these constraints is in the supplemental materials.

## Positive and negative descriptors

Consistent with expectations, descriptors that were rated as negative were used more frequently when describing vegans than when describing vegetarians, and negative descriptors were used more frequently when describing vegetarians than when describing omnivores. The most striking of these results is that negative characteristics comprised 22% of the descriptors participants provided when they thought of vegans, vs. 7% and 5% of the descriptors for vegetarians and omnivores respectively.

**Table 3. Mean percent of characteristics for each type of target.**

|  | Vegan | Vegetarian | Omnivore |
|---|---|---|---|
| Positive | 26.13[a] | 30.96[b] | 23.23[a] |
| Negative | 22.43[a] | 7.13[b] | 4.97[c] |
| Health | 15.44[a] | 15.44[a] | 5.80[b] |
| Environment | 9.76[a] | 9.87[a] | 4.20[b] |
| Animal rights | 1.21[a] | 0.45[b] | 0.28[b] |
| Lifestyle | 10.55[a] | 8.88[b] | 1.74[c] |
| Politics | 1.97[a] | 1.12[b] | 0.28[c] |
| Feminine | 2.15[a] | 1.96[a] | 0.05[b] |
| Deviance | 2.56[a] | 0.76[b] | 0.14[c] |
| Food | 21.01[a] | 34.15[b] | 35.69[b] |

Note: Within each row, means sharing a superscript were not significantly different at $p < .05$ or beyond.

Positive descriptors were used more frequently when referring to vegetarians than they were when referring to vegans and omnivores. Positive descriptors were used more frequently when describing vegans vs. omnivores, although the test of this difference did not reach conventional levels of significance ($p = .06$).

## Motives for adopting a veg*n diet

We coded descriptors in terms of whether they referred to the three most common motives people have for restricting meat from their diets: environmental concerns, health, and animal rights/welfare. Descriptors referring to environmental issues and health were used more frequently when describing both vegans and vegetarians than when describing omnivores. Descriptors referring to animal rights were used more often when describing vegans than when describing vegetarians and omnivores.

## Social identity descriptors

To determine the extent to which people think of veg*nism as a social identity, we examined differences in the use of four descriptors: lifestyle, deviance, politics, and femininity. The analyses of how frequently participants used descriptors referring to lifestyle, deviance, and politics found that all three groups differed significantly from one another. These characteristics were used more frequently when describing vegans than when describing vegetarians and were used more frequently for vegetarians than for omnivores. Descriptors referring to feminine sex roles were used more frequently when describing veg*ns than when describing omnivores. There was no significant difference on this measure between descriptions of vegans and vegetarians.

## Other descriptors

Descriptors rated as referring to food were used less frequently when describing vegans than when describing vegetarians and omnivores, between which there was no significant difference.

## Relationships between participants' diets and target descriptors

The previous analyses did not consider the possibility that the use of descriptors when referring to vegan, vegetarians, and omnivores would vary as a function of participants' diets. As

noted previously, the majority of participants were omnivores of some kind (omnivores or occasional omnivores, $n$ = 566), which means that the previous results represent the perceptions of omnivores more than they represent the perceptions of non-omnivores.

To examine the possibility that frequency of descriptions varied as a function of participants' diets, we regressed log-odds for each of the ten dimensions onto a continuous measure representing the extent to which people restricted meat from their diets. The level-1 model (word-level) was the same as that used in the previous analyses so that separate coefficients were estimated for vegans, vegetarians, and omnivores. Participants' restriction of meat was based on their reports of their diets and was coded as follows: omnivores, 0; occasional omnivores, 1; semi-vegetarians, 2; pescatarians, 3; lacto and lacto-ovo vegetarians, 4; and vegans, 5.

We used a continuous measure of meat restriction for technical and conceptual reasons. Technically, the covariance matrices were more stable when diet was represented as a continuous measure than when it was represented as a category (e.g., vegetarian or not). Conceptually, we thought of the underlying construct as a continuum more than a category. Moreover, the results of analyses based on categories (e.g., vegans and vegetarians vs. all others) led to the same conclusion as the analyses we present.

Relationships between meat restriction and evaluations were examined with the following model:

Word-level: $\eta_{ij}$ = $\beta_{1j}$ (vegan) + $\beta_{2j}$ (vegetarian) + $\beta_{3j}$ (omnivore)
Person-level: $\beta_{1j}$ = $\gamma_{10}$ + $\gamma_{11}$(restrict) + $\mu_{1j}$
$\beta_{2j}$ = $\gamma_{20}$ + $\gamma_{21}$(restrict) + $\mu_{2j}$
$\beta_{3j}$ = $\gamma_{30}$ + $\gamma_{31}$(restrict) + $\mu_{3j}$

The predictor, restrict, was entered uncentered. The hypotheses of interest (was how often a descriptor was used related to meat restriction) were tested at the person-level by tests of the $\gamma_{11}$, $\gamma_{21}$, and $\gamma_{31}$ coefficients.

The results of the analyses of all measured are summarized in S1 Table, which is available in the supplemental materials on the OSF site for this paper. This summary includes estimated values for the percent of descriptors for each dimension for each level of participants' meat restriction. Note that MLM estimates unstandardized coefficients, so the slopes represent the expected change for an outcome associated with a 1.0 increase in a predictor. Note also that the results of analyses of feminine descriptors are not presented in this table. The models for the analysis of this measure would not converge.

The two most striking sets of results were for the two global valanced descriptors, positive and negative. For descriptions of vegans and vegetarians, participants' meat restriction was positively related to how often positive descriptors were used: vegans, $\gamma_{11}$ = .196, $t$ = 4.05, $p <$ .001; vegetarians, $\gamma_{21}$ = .114, $t$ = 2.26, $p$ = .001;, and meat restriction was negatively related to how often negative descriptors were used: vegans, $\gamma_{11}$ = -.216, $t$ = 3.65, $p <$ .001; vegetarians, $\gamma_{21}$ = -.591, $t$ = 4.48, $p <$ .001. In contrast, for descriptions of omnivores, the reverse occurred: meat restriction was negatively related to how often positive descriptors were used: $\gamma_{31}$ = -.156, $t$ = 3.05, $p$ = .001;, and meat restriction was positively related to how often negative descriptors were used: $\gamma_{31}$ = .309, $t$ = 3.74, $p <$ .001. For evaluations of omnivores, meat restriction was positively related to how often descriptors related to deviance were used: $\gamma_{31}$ = .716, $t$ = 2.05, $p$ = .019. The results for positive and negative descriptors are depicted in Fig 1.

There were only a few relationships between participants' diets and the use of descriptors in the other categories. In terms of motives to adopt a veg*n diet, for descriptions of vegetarians and vegans, participants' meat restriction was positively related to how frequently descriptors related to the environment were used: vegans, $\gamma_{11}$ = .177, $t$ = 3.30, $p$ = .001; vegetarians, $\gamma_{21}$ = .102, $t$ = 2.03, $p$ = .042. For omnivore targets, participants' meat restriction was negatively related to how often descriptors related to health were used: $\gamma_{31}$ = -.223, $t$ = 2.06, $p$ = .042. For

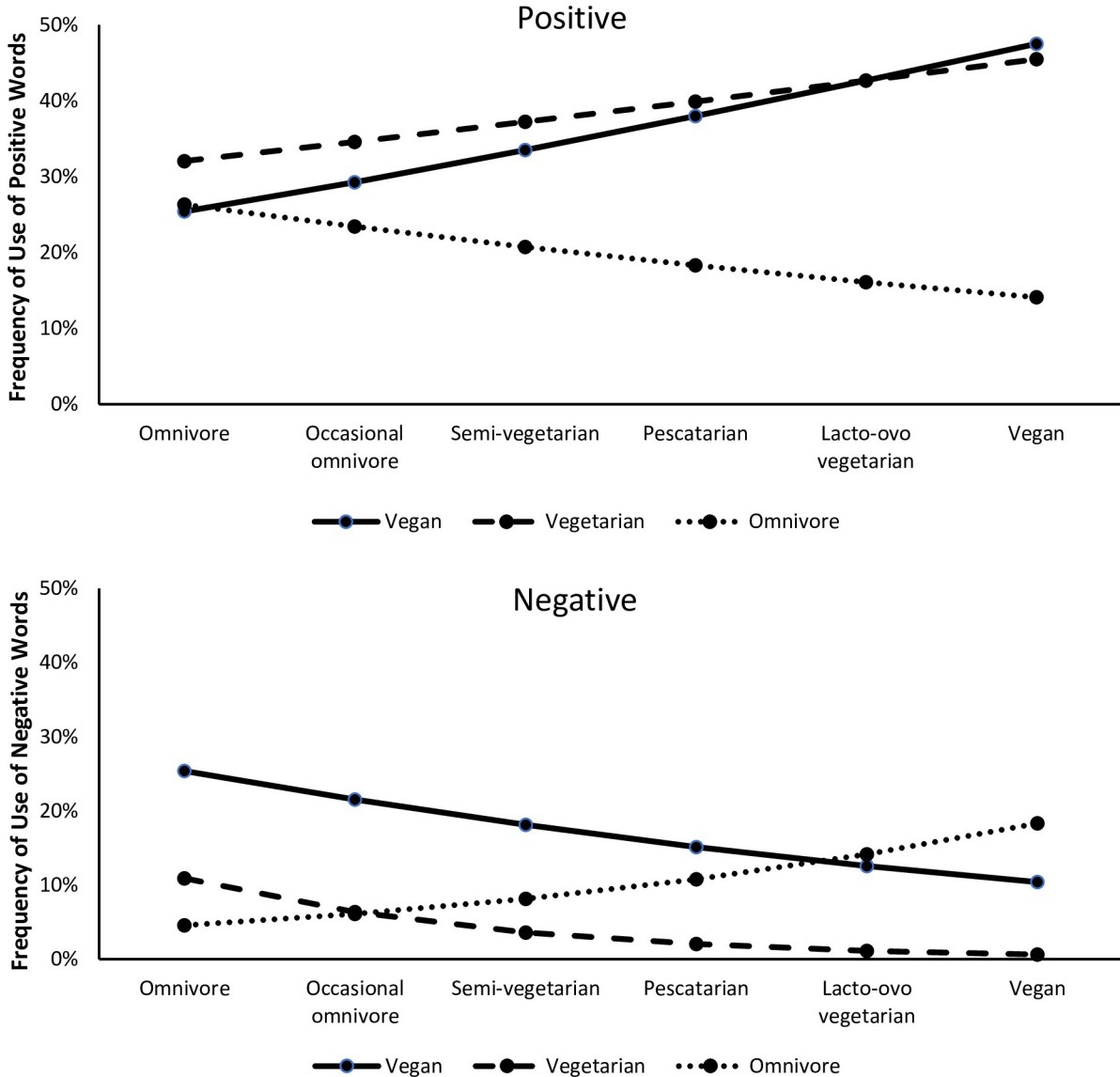

**Fig 1. Estimated frequency of use of words to describe vegan, vegetarian, and omnivore targets as a function of participants' dietary habits (x-axis).** Notes: (a) frequency of positive words. (b) frequency of negative words.

vegetarian targets, participants' meat restriction was negatively related to how frequently descriptors related to politics were used: $\gamma_{21} = -.559$, $t = 2.04$, $p = .042$.

## Discussion

The goal of the current study was to extend previous research by examining spontaneous verbal descriptions of people based solely on their dietary habits. Because veg*nism is considered to be a social identity that encompasses a variety of beliefs and attitudes, we predicted that people's spontaneous verbal descriptions would reflect veg*ns' motives for restricting their meat intake, as well as people's perceptions of the attitudes and beliefs associated with this social identity. Additionally, because veg*ns are a social minority, we believed that participants would use more negative descriptors to describe this group relative to omnivores–but this

would be moderated by participants' eating habits, which represented the degree to which they viewed target as members of out-groups.

## Valanced descriptors

As expected, negative descriptors were used more often when participants were describing vegans and vegetarians than they were used when describing omnivores, and negative descriptors were used more often when describing vegans than when describing vegetarians. More importantly, we also found that the use of negative descriptors for veg*ns was negatively related to the extent to which participants restricted meat from their diets, whereas the use of negative descriptors for omnivores was positively related to the extent to which participants restricted meat from their diets. Such differences are consistent with much of the previous research that has used close-ended response formats [14,16].

Positive descriptors were used more frequently when describing vegetarians than when describing vegans or omnivores. It is possible that the more frequent use of positive descriptors when describing vegetarians compared to vegans reflects relatively negative evaluations of vegans. This possibility was confirmed by an analysis that found that the difference between vegans and vegetarians in the use of positive descriptors disappeared after controlling for the use of negative descriptors. The constraint comparing vegans and vegetarians was not significant ($\chi^2 < 1$) after controlling for the use of negative descriptors.

We also found that the use of positive descriptors for veg*ns was positively related to the extent to which participants restricted meat from their diets, whereas the use of positive descriptors for omnivores was negatively related to the extent to which participants restricted meat from their diets. We believe these results reflect the operation of a simple principle: People evaluate others who are more similar to themselves more positively and less negatively than others who are dissimilar to themselves. This is an extension of the well-established finding that similarity leads to attraction [27], and it is consistent with research demonstrating that friends and lovers and more likely to have similar diets than expected by chance [32].

Regardless, it is difficult to discuss how well these findings agree or disagree with previous research because very few studies have examined differences in positive perceptions of these three target groups simultaneously. Moreover, the results of previous studies have been mixed in terms of positive evaluations of vegans and vegetarians.

## Motive-related descriptors

Consistent with the motives veg*ns report for restricting meat, descriptors referring to environmental issues and health were used more frequently when describing both vegans and vegetarians than when describing omnivores. Animal rights, the other common motive for restricting meat from one's diet, was used more often when describing vegans than when describing vegetarians and omnivores. These results suggest that the cognitive representations of the reasons why people follow veg*ns diets (i.e., the frequency of the use of descriptors) correspond to the actual motives of veg*ns.

Moreover, the relative use of some of these descriptors varied as a function of participants' restriction of meat from their diets. The use of health-related descriptors for omnivores was negatively related to participants' meat restriction. This suggests that the more people restrict meat from their diets the less healthy they perceive an omnivorous diet to be, a factor which likely plays a role in their own dietary decisions.

The use of environmental-related descriptors for veg*ns was positively related to participants' meat restriction. This suggests that the more people restrict meat from their diets the more they associate concerns for the environment with a veg*n diet. This may simply reflect

the fact that environmental concerns are important reasons people adopt a veg*n diet, and the more people restrict meat from their diets the more they are aware of this, if only because it may apply to them more strongly. In contrast, environmental concerns are not particularly salient for omnivores, and veg*ns are as aware of this as omnivores are.

## Social identity descriptors

Participants' use of descriptors also provided support for the contention that veg*nism is a social identity. Descriptors referring to deviance, lifestyle, and politics were used more frequently when referring to vegans than when describing vegetarians and were used more frequently when describing both vegans and vegetarians than when describing omnivores. Also, feminine descriptors were used more frequently when describing veg*ns than when describing omnivores. Moreover, none of the frequencies of the use of these descriptors was related to the diets of perceivers, suggesting that people who follow different diets conceptualize veg*nism as a social identity similarly. Because food choice is a domain within which people express their ideals and identities, it connects communities of individuals who share similar food-choice patterns. By extension, these values and beliefs link food choices to both personal and social identity [2,10].

With respect to deviance, we found that descriptors coded as representing deviance were more likely to be rated as negative and were less likely to be rated as positive than descriptors not coded as deviance (both $p$s < .0001). Moreover, the use of deviance-related descriptors was related to participants' restriction of meat from their diets for only omnivore targets. The more participants restricted meat from their diets, the more likely they were to use deviance related descriptors when describing omnivores. This may reflect a tendency for those who restrict their meat intake to establish a new norm (meat restriction), which could entail thinking of omnivores as deviants in some ways.

## Other descriptors

We also found that food-related descriptors were used less frequently when describing vegans than when describing vegetarians or omnivores, which did not differ. Speculatively speaking, it is possible that this difference reflects the fact that vegans eat fewer types of foods than either vegetarians or omnivores. By extension, the cognitive representation of vegans may not include as many food-related entities as the representations of vegetarians and omnivores. Veganism may be thought of more in terms of what vegans do not eat, rather than in terms of what they eat. Moreover, the relative use of food-related descriptors was not related to the diets of perceivers, suggesting that excluding food-related terms from the conceptualization of vegans does not vary as a function of how much people restrict meat from their diets.

## Strengths and limitations

The value of any study is limited by the constructs that were measured and the sample that was studied. In the present study, we used spontaneous descriptions of vegans, vegetarians, and omnivores and coded these descriptors using 10 dimensions based on previous research and theory. When we developed the coding scheme, we had numerous discussions, and these discussions included our initial raters (who were drawn from the same population as our participants), and there did not seem to be enough descriptors to warrant including an additional dimension. It is possible that if we had asked participants to provide more than four descriptors, we would have had a basis to add to the 10 dimensions we used.

There is also the issue of the low relative frequency (e.g., less than 10%) with which some descriptor for some dimensions were used. Although we found differences in how often such

descriptors were used when describing our three targets, such low relative frequencies suggest that some dimensions were not that salient. Moreover, it is possible that these lower frequencies made it difficult to find relationships between participants' diets and their ratings–too many zeroes as outcomes. Such issues need to be addressed in future studies perhaps by allowing perceivers to list more descriptors.

Participants were a convenience sample of US collegians, and as such they represent a WEIRD (i.e., White, Educated, Industrialized, Rich, and Democratic) sample [33]. Putting this aside for the moment, the percent of vegetarians in our sample (6.0% vegans, lacto-, and lacto-ovo vegetarians) is comparable to the percent of vegetarians in the US [12]. We will not speculate about how the specific characteristics of our sample may have influenced our results because we have no basis to do so.

Nevertheless, it is important to note that the vast majority of research on vegetarianism has been done in societies and cultures (primarily Western) in which vegetarians are a minority, less than 10% of the population [34]. Moreover, as discussed by Leahy et al., these are primarily "vegetarians of choice" vs. "vegetarians of necessity." Our participants lived in a country in which meat is available to virtually all members of society.

In contrast, many of the world's vegetarians are vegetarians of necessity because meat is not accessible (e.g., not available, too expensive). For members of such societies, being a vegetarian is probably not a salient identity (almost everyone is a vegetarian), and so following a vegetarian diet may not be a particularly important influence on person perception. The influence of diet on person perception needs to be understood within the social context within which people live.

## Conclusions

The present results suggest that spontaneous verbal descriptions are a useful way to study perceptions of veg*ns and other dietary groups. We found reliable differences in the spontaneous verbal descriptors used to define dietary groups and that how people perceived others varied as a function of people's own dietary habits. These findings suggest that it is important to consider both the dietary habits of the target and perceiver when assessing interpersonal perceptions based on eating habits.

The findings of this study as well as others [14,16], suggest that veg*ns may be viewed negatively, especially by those with different dietary habits (i.e., omnivores). Because veg*n dietary habits, attitudes, and beliefs challenge traditional societal norms, they may threaten others who do not hold similar views. As a result, those who adopt veg*n diets may experience disrespect, derogation, and stigmatization, leading to negative social experiences that may pose threats to their psychological well-being [e.g., 35].

Our results also suggest that veg*ns think of omnivores more negatively than they think of other veg*ns. The vast majority of research on perceptions of people as a function of their diets has focused on how veg*ns are derogated by non-veg*ns. In fact, many studies do not have control groups of omnivorous targets to provide a basis to see how omnivores are perceived. Our results suggest that there may be more "mutual hostility" between veg*ns and omnivores than previous research has suggested. Answering such questions will require research specifically designed to do so.

## Acknowledgments

We thank Grace Li, Abby Carlson, Rachel Akers, Katie Kelly, and Kristen Fritzeen for their help coding the responses.

## Author Contributions

**Conceptualization:** John B. Nezlek, Catherine A. Forestell, Harini Krishnamurti.

**Data curation:** John B. Nezlek, Catherine A. Forestell, Harini Krishnamurti.

**Formal analysis:** John B. Nezlek, Catherine A. Forestell.

**Methodology:** Harini Krishnamurti.

**Writing – original draft:** John B. Nezlek, Catherine A. Forestell, Harini Krishnamurti.

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
