## [Decision Letter · Decision Letter 0]

22 May 2023

PONE-D-23-05809Spontaneous verbal descriptions of vegans, non-vegan vegetarians, and omnivores and relationships between these descriptions and perceivers’ dietsPLOS ONE

Dear Dr. Nezlek,

Thank you for submitting your manuscript to PLOS ONE. After careful consideration, we feel that it has merit but does not fully meet PLOS ONE’s publication criteria as it currently stands. Therefore, we invite you to submit a revised version of the manuscript that addresses the points raised during the review process.

Both reviewers agree that you present an interesting study and I feel the same. Regarding necessary improvements, both reviewers agree that you should broaden your introduction and include more relevant theory and research; that your description of participants’ diets needs improvement (e.g., “restriction of meat” is inaccurate and confusing); that the generalizability of your findings needs addressing and that your conclusions require some reworking. In addition, Reviewer 1 raised important concerns about the rating of your descriptors. I would encourage you to take particular care of these points in your revision.

There are some additional points that I would like you to address/consider:

A) In your Data Availability Statement, please state where the data can be found.B) Your analyses address relative proportions, but throughout your paper you often talk about means, which I found confusing. Please be more consistent. Similarly, is “within persons” (238) the same as “within j participants” (242)? Again, please check for consistency throughout.C) Many readers will not be familiar with the statistical models you applied. Therefore, please address the meaning of your results (tables) in greater detail. What can we conceive as the population that the estimates (Table 2) address? What does variance in Table 2 mean (e.g., 20x larger for animal rights than politics). What is the unit for variance? What do the variances μ1j, μ2j, and μ3j (293) capture?D) Please consider whether key results in Table 4 might be better presented as a graph (potentially with a Supplementary Table for additional details). ==============================

We look forward to receiving your revised manuscript.

Kind regards,

Johannes Hönekopp

Academic Editor

PLOS ONE

Reviewers' comments:

Reviewer's Responses to Questions

**Comments to the Author**

1. Is the manuscript technically sound, and do the data support the conclusions?

Reviewer #1: Partly

Reviewer #2: Yes

2. Has the statistical analysis been performed appropriately and rigorously? 

Reviewer #1: Yes

Reviewer #2: Yes

3. Have the authors made all data underlying the findings in their manuscript fully available?

Reviewer #1: Yes

Reviewer #2: Yes

4. Is the manuscript presented in an intelligible fashion and written in standard English?

Reviewer #1: Yes

Reviewer #2: Yes

5. Review Comments to the Author

Reviewer #1: It is an interesting and timely study aiming to analyse the attitudes towards veg*ns and omnivores through analyses of spontaneous descriptions made by participants following various diets. The study found that vegans were viewed more negatively than vegetarians, who were viewed more negatively than omnivores, but this was moderated by the extent to which participants restricted meat from their diets. Although the method is quite unique and results are presented in a clear and orderly manner, the manuscript has some significant limitations (see comments below).

My main concern is to what extent the study shows the attitude to the specific word/terms of a few undergraduate students (raters) rather than perceptions of veg*ns and omnivores by general populations. For example, the trait “free-spirited” may be rated as negative by a conservative or deeply religious person. Moreover, rating adjectives such as “caring”, “flowery”, or “soft” as feminine is very controversial nowadays. It is difficult to judge how serious the bias can be as you presented in table 1 only examples of descriptions.

The abstract is quite uncomfortable to follow as it presents a lengthy summary of results and lacks clear conclusions.

Lines 62-82: It would be helpful to present some examples of what it means to be viewed more positively or negatively (in what respect?). Statements such as “positively in some ways” (line 72 and 79) is very ambiguous.

The review of the literature is very limited. It only refers to four previous works. The major part of the Introduction section presents the description of the methods. It resembles the structure of an introduction to some thesis rather than the research manuscript.

Moreover, numerous papers present positive attitudes toward veg*ns (eg. Chin, M. G., Fisak Jr, B., & Sims, V. K. (2002). Development of the attitudes toward vegetarians scale. Anthrozoös, 15(4), 332-342; Judge, M., & Wilson, M. S. (2019). A dual‐process motivational model of attitudes towards vegetarians and vegans. European Journal of Social Psychology, 49(1), 169-178; Ruby, M. B., & Heine, S. J. (2011). Meat, morals, and masculinity. Appetite, 56(2), 447-450.). The contradictory results of both hypotheses and findings are not appropriately discussed in the manuscript.

Line 123-124: the concept of “vegetarianism as a social identity” is only briefly mentioned in the description of the methodology. It would be useful to present it more broadly. Or any other psychological/sociological concept which forms this study’s bases.

Line 147-148; 153-156; 157-159: References?

Methods: Why didn’t you ask participants to present negative, positive and neutral words? You would have immediately received the valence of trait and other terms without the bias of the raters.

Lines 340-342: I think the lack of such analyses is due to the logical assumption that the veg*ns wouldn’t have a negative attitude to veg*ns. Please, comment.

Page 23: Is it possible to draw reliable conclusions from a correlation between 4 or 10 participant groups with a group of 480?

Discussion: The discussion thoroughly and clearly presents the results but needs more theoretical analyses of the findings. The manuscript only briefly refers to some psychological concepts, such as social perception, attitudes towards minorities, etc., but does not discuss results in the view of any theory.

Page 28: “It is possible that if we had asked participants to provide more than four descriptors, we would have had a basis to add to the ten dimensions we used.” Isn’t that obvious?

Page 29: “In contrast, many of the world’s vegetarians are vegetarians of necessity because meat is not accessible (e.g., not available, too expensive). For members of such societies, being a vegetarian is probably not a salient identity (almost everyone is a vegetarian), and so following a vegetarian diet may not be a particularly important influence on person perception.” Is it really a conclusion from that study? How does it refer to any of the findings or the overall topic?

Reviewer #2: This is an interesting article and brings a new perspective related to difference in people's attitudes based on their diets. It also generates some valuable insights.

The following are my concerns in relation to the study and how it is presented:

• There doesn't seem to be any theoretical background - some theories are briefly mentioned in the Discussion sections but are missing in the justification for the study;

• The gap in previous research and studies is not explicitly stated;

• There is no discussion about how valid or applicable the findings are given the nature of the sample used;

• The paper could benefit from better engaging with the issue of why such results were obtained, that is, what explains the findings;

• Some terms need to be explained for the wider audience, namely WEIRD sample, woke, Peta (I assume it's PETA);

• There is no indication when the study was conducted;

• It seems strange that a different range of diets is used to describe the participants' diets compared to the evaluated diets (e.g. Table 4) - why was such a choice made?

• How can there be no dimension scores for one quarter of the descriptors (line 231)? It needs better wording/explanation;

• Using sentences which start with "Note" does not seem appropriate in an academic article;

• There should be more consistency in the use of the term vegetarian or a definition given - at the moment, it is described both as "non-vegan vegetarian" and "lacto-ovo vegetarian" as well as simply vegetarian; similarly, what is meant by "meat restrictors"?

• It is best to have the limitations of the study outside of the Conclusion;

• The Conclusion should be improved as it does not do justice of the research in terms of presenting its findings and implications.

6. PLOS authors have the option to publish the peer review history of their article (what does this mean?). If published, this will include your full peer review and any attached files.

Reviewer #1: No

Reviewer #2: No

---

## [Author Response · Author response to Decision Letter 0]

22 Sep 2023

Both reviewers agree that you present an interesting study and I feel the same. Regarding necessary improvements, both reviewers agree that you should broaden your introduction and include more relevant theory and research; that your description of participants’ diets needs improvement (e.g., “restriction of meat” is inaccurate and confusing); that the generalizability of your findings needs addressing and that your conclusions require some reworking. In addition, Reviewer 1 raised important concerns about the rating of your descriptors. I would encourage you to take particular care of these points in your revision.

There are some additional points that I would like you to address/consider:

A) In your Data Availability Statement, please state where the data can be found.

This was done.

B) Your analyses address relative proportions, but throughout your paper you often talk about means, which I found confusing. Please be more consistent. Similarly, is “within persons” (238) the same as “within j participants” (242)? Again, please check for consistency throughout.

Reply: Point well-taken. The analyses estimate log-odds which are then transformed into percentages, and we have tried to make clearer exactly what the numbers represent. When describing the analyses, we changed within-persons to word-level and between person to person-level to make clearer what our results represent. We also made numerous other changes to help readers who might not be that familiar with multilevel modeling.

C) Many readers will not be familiar with the statistical models you applied. Therefore, please address the meaning of your results (tables) in greater detail. 

• What can we conceive as the population that the estimates (Table 2) address? 

• What does variance in Table 2 mean (e.g., 20x larger for animal rights than politics). 

• What is the unit for variance? What do the variances μ1j, μ2j, and μ3j (293) capture?

Reply: We agree that the descriptions of the analyses and the results were inadequate, particularly given that many readers will probably not be that familiar with the types of analyses we did. So, we have provided more detail in numerous places, including a brief discussion of why we used multilevel modeling. 

In terms of your specific questions, the values in Table 2 are estimates of the population parameters, i.e., the mean percent of words that received different evaluations. These are based on the log-odds estimated by the analyses, something we make clearer in the revision. The variance estimates are the person-level variances of the log-odds, i.e., estimates of the random effects. The same can be said of the data presented in Table 3.

D) Please consider whether key results in Table 4 might be better presented as a graph (potentially with a Supplementary Table for additional details). 

Reply: Good idea. We have included a figure depicting the key results of Table 4 (positive and negative evaluations as a function of diet), and we moved Table 4 to the Supplemental materials. 

Reviewer #1

It is an interesting and timely study aiming to analyze the attitudes towards veg*ns and omnivores through analyses of spontaneous descriptions made by participants following various diets. The study found that vegans were viewed more negatively than vegetarians, who were viewed more negatively than omnivores, but this was moderated by the extent to which participants restricted meat from their diets. Although the method is quite unique and results are presented in a clear and orderly manner, the manuscript has some significant limitations (see comments below).

1. My main concern is to what extent the study shows the attitude to the specific word/terms of a few undergraduate students (raters) rather than perceptions of veg*ns and omnivores by general populations. For example, the trait “free-spirited” may be rated as negative by a conservative or deeply religious person. Moreover, rating adjectives such as “caring”, “flowery”, or “soft” as feminine is very controversial nowadays. It is difficult to judge how serious the bias can be as you presented in table 1 only examples of descriptions.

Response: The reviewer is correct in noting that our findings are limited by the fact that the descriptors were generated and rated by undergraduate students, and we consider this issue in the discussion section. In terms of the ratings themselves, all ratings were reviewed by two tenured professors, and all of the ratings that were analyzed were approved by them. So, we think that the ratings are valid for our sample. Would a different sample have provided different terms, perhaps, and would these different terms have led us to construct a different set of dimensions, perhaps? Nevertheless, we were not concerned with the words per se; rather, we were concerned with what the words represented, i.e., the dimensions. 

Like other qualitative studies, we had trained coders, and we chose coders who were university students from the same population as our sample because their interpretation of the meaning and intention of the words would be more valid than ratings made by raters drawn from another population. Moreover, all coders were blind to the dietary habit of participants. It is possible that a term such as “free-spirited” could be negatively evaluated by someone, but such possibilities are beyond the scope of this study. Our coders uniformly classified free-spirited as positive. Nevertheless, only one person used free-spirited, and given this, we have removed it as an exemplar. In terms of the exemplars listed in Table 1, we have edited them so that they represent the more commonly used exemplars for a dimension. A list of all terms is available in the online supplemental materials.

We recognize that it is not appropriate to stereotype women as being “caring”, “flowery”, or “soft”, but these terms are consistent with traditional feminine gender roles, and in the present study, such words were used more frequently to describe vegetarians than omnivores by all participants, regardless of their diet.

2. The abstract is quite uncomfortable to follow as it presents a lengthy summary of results and lacks clear conclusions.

Response: We have shortened and clarified the abstract based on the Reviewer’s feedback.

3. Lines 62-82: It would be helpful to present some examples of what it means to be viewed more positively or negatively (in what respect?). Statements such as “positively in some ways” (line 72 and 79) is very ambiguous.

Response: We have provided examples of how vegetarians reported that omnivores viewed them positively from Nezlek et al., 2023.

4. The review of the literature is very limited. It only refers to four previous works. The major part of the Introduction section presents the description of the methods. It resembles the structure of an introduction to some thesis rather than the research manuscript. Moreover, numerous papers present positive attitudes toward veg*ns (eg. Chin, M. G., Fisak Jr, B., & Sims, V. K. (2002). Development of the attitudes toward vegetarians scale. Anthrozoös, 15(4), 332-342; Judge, M., & Wilson, M. S. (2019). A dual‐process motivational model of attitudes towards vegetarians and vegans. European Journal of Social Psychology, 49(1), 169-178; Ruby, M. B., & Heine, S. J. (2011). Meat, morals, and masculinity. Appetite, 56(2), 447-450.). The contradictory results of both hypotheses and findings are not appropriately discussed in the manuscript.

Response: We have included more background in the introduction. 

5. Line 123-124: the concept of “vegetarianism as a social identity” is only briefly mentioned in the description of the methodology. It would be useful to present it more broadly. Or any other psychological/sociological concept which forms this study’s basis.

Response: A more detailed discussion of vegetarianism as a social identity is provided in the first two paragraphs of the introduction.

6. Line 147-148; 153-156; 157-159: References?

Response: Appropriate references have been added.

Methods: 

7. Why didn’t you ask participants to present negative, positive and neutral words? You would have immediately received the valence of trait and other terms without the bias of the raters.

Response: The goal of the present study was for participants to provide spontaneous verbal descriptions to allow us to assess the degree to which these descriptions were positive and negative for vegans, vegetarians, and omnivores. It would have undermined the purpose of the study if we had defined the valence of the terms.

8. Lines 340-342: I think the lack of such analyses is due to the logical assumption that the veg*ns wouldn’t have a negative attitude to veg*ns. Please, comment.

Response: The previous analyses that we refer to on lines 340-342 is referring to our own analyses (not those of others). The goal of the present paper was to assess the words participants produced to describe vegans, vegetarians and omnivores as a function of their own dietary habits. We hypothesized that people would have more negative views toward dietary outgroups. Therefore, although the first analysis that we reported did not “consider the possibility that the use of descriptors when referring to vegan, vegetarians, and omnivores would vary as a function of participants’ diets”, the second analyses took participants’ dietary habit into account. 

9. Page 23: Is it possible to draw reliable conclusions from a correlation between 4 or 10 participant groups with a group of 480?

Response: It is not clear to us what the reviewer is referring to here. To what does the phrase “a correlation between 4 or 10 participants groups” refer? We had only one group of participants. Regardless, the MLM analyses we used incorporate what is called “precision weighting,” which takes into account differences in the number of observations across units of analysis. Similarly, in logistical regression, tests of significance take into account the distribution of responses. 

Discussion: 

10. The discussion thoroughly and clearly presents the results but needs more theoretical analyses of the findings. The manuscript only briefly refers to some psychological concepts, such as social perception, attitudes towards minorities, etc., but does not discuss results in the view of any theory.

Response: Because we have provided more theoretical background in the introduction, we have included a paragraph at the beginning of the discussion to remind the reader of the study goals and hypotheses, which are built on the foundation of Social Identity Theory.

11. Page 28: “It is possible that if we had asked participants to provide more than four descriptors, we would have had a basis to add to the ten dimensions we used.” Isn’t that obvious?

Response: We do not think it is obvious. More descriptors would not necessarily provide a basis for more dimensions. The fifth (and beyond) descriptors might be able to be subsumed under the 10 dimensions we used.

12. Page 29: “In contrast, many of the world’s vegetarians are vegetarians of necessity because meat is not accessible (e.g., not available, too expensive). For members of such societies, being a vegetarian is probably not a salient identity (almost everyone is a vegetarian), and so following a vegetarian diet may not be a particularly important influence on person perception.” Is it really a conclusion from that study? How does it refer to any of the findings or the overall topic?

Response: We have separated the section about limitations as strengths from the Conclusions of the study.

 

Reviewer #2: 

This is an interesting article and brings a new perspective related to difference in people's attitudes based on their diets. It also generates some valuable insights. The following are my concerns in relation to the study and how it is presented:

1. There doesn't seem to be any theoretical background - some theories are briefly mentioned in the Discussion sections but are missing in the justification for the study.

Response: We have provided more theoretical background in the Introduction.

2. The gap in previous research and studies is not explicitly stated.

Response: There are numerous gaps in the existing literature, and we have tried to make them clear. Most directly relevant to the present study, we are aware of only one study that has examined spontaneous verbal descriptions as a function of diet, Minson and Monsin (2012), a study with a sample of 47 in which descriptions of omnivores were not collected. We should add that in many studies on this topic, perceptions of omnivores are not collected.

3. There is no discussion about how valid or applicable the findings are given the nature of the sample used.

Response: This is discussed in the revised section labeled Strengths and Limitations.

4. The paper could benefit from better engaging with the issue of why such results were obtained, that is, what explains the findings. 

Response: The discussion has separate section in which we discuss the results in terms of word valence, motives, social identity, and other descriptors that did not fit into those just mentioned. These discussions include why we believe we found what we found. 

5. Some terms need to be explained for the wider audience, namely WEIRD sample, woke, Peta (I assume it's PETA);

Response: These definitions are now included in the manuscript.

6. There is no indication when the study was conducted;

Response: Good point. As noted in the revision, the data were collected between September and December, 2020.

7. It seems strange that a different range of diets is used to describe the participants' diets compared to the evaluated diets (e.g. Table 4) - why was such a choice made?

Response: We wanted to use a continuous scale to describe participants’ dietary habits, however it was not reasonable to ask participants to provide three descriptors for six dietary groups – especially since they may be less familiar with groups such as occasional omnivores, or pescatarians.

8. How can there be no dimension scores for one quarter of the descriptors (line 231)? It needs better wording/explanation.

Response: As described in the paper, approximately 25% words could not be rated as reflecting any of our ten dimensions, for example, common. There is not much more we can say about this.

9. Using sentences which start with "Note" does not seem appropriate in an academic article.

Response: We have changed some of these, but not all. We are unaware of any style guidelines that recommend against this. In fact, in Strunk and White’s classic guide, The Elements of Style (which is an academic text to be certain), numerous sentences (seven to be precise) start with Note. They used this construction as we did, to draw the reader’s attention.

10. There should be more consistency in the use of the term vegetarian or a definition given - at the moment, it is described both as "non-vegan vegetarian" and "lacto-ovo vegetarian" as well as simply vegetarian; similarly, what is meant by "meat restrictors"?

Response: This refers to those who restrict their meat intake;, vegans, vegetarians, pescatarians, semi-vegetarians, and flexitarians. Rather than list out all of these groups we refer to them collectively. In the manuscript we have removed this term and now refer to “those who restricted their meat intake” to clarify what we mean.

11. It is best to have the limitations of the study outside of the Conclusion;

Response: We have separated the limitations and Conclusions of the study

12. The Conclusion should be improved as it does not do justice of the research in terms of presenting its findings and implications.

Response: We have edited the conclusion to improve the presentation of the findings and implications of this study.

---

## [Decision Letter · Decision Letter 1]

23 Oct 2023

Spontaneous verbal descriptions of vegans, non-vegan vegetarians, and omnivores and relationships between these descriptions and perceivers’ diets

PONE-D-23-05809R1

Dear Dr. Nezlek,

We’re pleased to inform you that your manuscript has been judged scientifically suitable for publication and will be formally accepted for publication once it meets all outstanding technical requirements.

Kind regards,

Johannes Hönekopp

Academic Editor

PLOS ONE

Additional Editor Comments (optional):

Reviewers' comments:

Reviewer's Responses to Questions

**Comments to the Author**

1. If the authors have adequately addressed your comments raised in a previous round of review and you feel that this manuscript is now acceptable for publication, you may indicate that here to bypass the “Comments to the Author” section, enter your conflict of interest statement in the “Confidential to Editor” section, and submit your "Accept" recommendation.

Reviewer #1: All comments have been addressed

2. Is the manuscript technically sound, and do the data support the conclusions?

Reviewer #1: Yes

3. Has the statistical analysis been performed appropriately and rigorously? 

Reviewer #1: Yes

4. Have the authors made all data underlying the findings in their manuscript fully available?

Reviewer #1: Yes

5. Is the manuscript presented in an intelligible fashion and written in standard English?

Reviewer #1: Yes

6. Review Comments to the Author

Reviewer #1: Thank you for answering my questions and revising the manuscript. I have no further comments. Best regards!

7. PLOS authors have the option to publish the peer review history of their article (what does this mean?). If published, this will include your full peer review and any attached files.

Reviewer #1: **Yes: **Klaudia Modlinska

---

## [Editor Report · Acceptance letter]

31 Oct 2023

PONE-D-23-05809R1 

Spontaneous verbal descriptions
of vegans, non-vegan vegetarians, and omnivores and
relationships between these descriptions and perceivers’ diets 

Dear Dr. Nezlek:

I'm pleased to inform you that your manuscript has been deemed suitable for publication in PLOS ONE. Congratulations! Your manuscript is now with our production department. 

Kind regards, 

on behalf of

Dr. Johannes Hönekopp 

Academic Editor

PLOS ONE